# Effect of Dietary Inulin Supplementation on Growth Performance, Carcass Traits, and Meat Quality in Growing–Finishing Pigs

**DOI:** 10.3390/ani9100840

**Published:** 2019-10-21

**Authors:** Weikang Wang, Daiwen Chen, Bing Yu, Zhiqing Huang, Yuheng Luo, Ping Zheng, Xiangbin Mao, Jie Yu, Junqiu Luo, Jun He

**Affiliations:** Institute of Animal Nutrition, Sichuan Agricultural University, and Key Laboratory of Animal Disease-Resistance Nutrition, Ministry of Education, Chengdu 611130, China; wwk335577@163.com (W.W.); chendwz@sicau.edu.cn (D.C.); ybingtian@163.com (B.Y.); zqhuang@sicau.edu.cn (Z.H.); luoluo212@126.com (Y.L.); zpind05@163.com (P.Z.); acatmxb2003@163.com (X.M.); yujie@sicau.edu.cn (J.Y.); junqluo2018@tom.com (J.L.)

**Keywords:** inulin, meat quality, carcass traits, pigs, metabolism

## Abstract

**Simple Summary:**

Dietary fiber has attracted considerable research interest worldwide. Inulin is a critical soluble dietary fiber. This study investigated the effects of dietary inulin supplementation on the growth performance and meat quality in pigs, which provided novel insights into the application of inulin for the livestock industry.

**Abstract:**

Inulin is one of the commercially feasible dietary fibers that has been implicated in regulating the gut health and metabolism of animals. This experiment was conducted to investigate the effect of dietary inulin supplementation on growth performance and meat quality in growing–finishing pigs. Thirty-six Duroc × Landrace × Yorkshire White growing barrows (22.0 ± 1.0 kg) were randomly allocated to two dietary treatments consisting of a basal control diet (CON) or basal diet supplemented with 0.5% inulin (INU). Results showed that inulin supplementation tended to increase the average daily gain (ADG) at the fattening stage (0.05 < *p* < 0.10). Inulin significantly increased the dressing percentage (*p* < 0.05) and tended to increase the loin-eye area. The serum concentrations of insulin and IGF-I were significantly higher (*p* < 0.05) in the INU group than in the CON group. Moreover, inulin supplementation significantly elevated the expression level of myosin heavy chain II b (*MyHC IIb*) in the longissimus dorsi (*p* < 0.05). Inulin significantly upregulated the expression of mammalian rapamycin target protein (*mTOR*) but decreased (*p* < 0.05) the expression level of muscle-specific ubiquitin ligase *MuRF-1*. These results show the beneficial effect of inulin supplementation on the growth performance and carcass traits in growing–finishing pigs, and will also facilitate the application of inulin in swine production.

## 1. Introduction

Inulin is a group of naturally occurring polysaccharides belonging to a class of dietary fiber known as fructans [1]. Inulin exists in a wide variety of fruits and vegetables, such as bananas, asparagus, leeks, and onions. However, most industrially produced inulin is extracted from chicory (*Compositae* family) and Jerusalem artichoke (*Helianthus tuberosus*) since they are extremely abundant in fructans. As an attractive dietary fiber, inulin cannot be hydrolyzed by mammal digestive enzymes and absorbed in the small intestine, but it can be partially hydrolyzed and fermented by intestinal microflora [2].

Previous studies indicated that inulin plays a critical role in maintaining gut health in human and mammalian animals, such as mice and pigs [3,4,5]. For instance, inulin was found to promote the highest villus height and the ratio of villus height/crypt depth in the jejunum and ileum [6]. Moreover, fermentation of inulin by intestinal bacteria produces a large number of short-chain fatty acids (SCFAs), such as the acetate, propionate, and butyrate, which can stimulate the production and secretion of the mucous layer covering the mucosal surface of the gastrointestinal tract [1]. This mucous layer is composed of high molecular glycoproteins and is considered to be the first line of defense against invasion [7]. Butyrate was found to serve as the primary energy source for colonocytes and also protect against colorectal cancer and inflammation [8]. In addition to SCFAs/butyrate beneficial effects, dietary fiber-derived SCFAs can also participate in the regulation of metabolisms. Mice fed a butyrate-enriched high-fat diet have increased thermogenesis and energy expenditure and are resistant to obesity [9]. The propionate is classically described as an efficient hepatic gluconeogenic substrate, but it also serves as a gluconeogenic substrate in the intestine before reaching the liver [10]. Moreover, studies on humans and animals indicated that dietary fibers can improve glucose metabolism via increasing insulin sensitivity [11]. Both these reports suggested beneficial roles of dietary fiber in the regulation of gut health and metabolism.

Currently, consumer demand for high-quality meat is increasing in most countries. Factors such as swine genetics, pre-slaughter handling, harvest, and pork carcass chilling affect the pork quality [12,13]. However, accumulating evidence shows that manipulating the nutrient composition of swine diets, such as adding dietary fiber, may enhance the meat quality traits [14]. Previous studies have indicated that dietary fibers affect growth performance and meat quality in a variety of animal species [15]. Zeng et al. reported that a mulberry leaf diet decreased cooking loss and drip loss, and elevated the expression level of *MyHC I* and *MyHC IIa* mRNA [16]. Moreover, increasing the fiber level in the diet was found to elevate the dressing percentage and decrease the intramuscular fat content in pigs [17]. As an attractive source of dietary fiber, inulin has been previously implicated in maintaining gut health and regulating metabolism [18]. However, little is known about its influence on meat quality. This study was conducted to explore the effects of dietary inulin supplementation on growth performance, blood metabolic parameters, carcass traits, and meat quality in growing–finishing pigs.

## 2. Materials and Methods

All procedures of animal experiments were carried out based on protocols approved by the Animal Care Advisory Committee of Sichuan Agricultural University.

### 2.1. Experimental Design

Thirty-six finishing pigs (Duroc × Landrace × Yorkshire, Duroc is the terminal boar, Landrace is the male parent, and Yorkshire is the female parent. Duroc is from the United States, Landrace is from Denmark, and Yorkshire is from North England.) with an average initial body weight of 22.0 ± 1.0 kg (70 days of age) were selected from the same herd and randomly allotted to two dietary treatments with six replications of three pigs (barrows) per replicate pen. The experimental diet was formulated based on nutrient requirements established by the National Research Council [19] (the diets were fed in three phases: 1–32 d, 32–70 d, and 70–96 d). The two treatments were a basal control diet (CON) or a basal diet with 5 g/kg of added inulin (99.5% basal diet + 0.5% inulin). Inulin with a 99.5% purity (extracted from chicory) was purchased from Beijing Zhongtaihe Technology Co., LTD. There were no discrepancies for other nutrient components. The ingredients and nutrient levels of the experimental diets are shown in Table 1. Pigs were housed in a totally enclosed building with galvanized slotted floors with 2.00 × 2.00 m pens during the experiment period. The building was naturally ventilated (windows) and was not environmentally controlled. Water was provided ad libitum. The animal trial lasted for 96 d, and pigs were handfed three times/d (8:30 am, 14:30 pm, and 20:30 pm) in groove feeders to make sure fresh feed was available ad libitum. Feed distribution of the pigs was determined daily throughout the trial on a pen basis. Average daily body weight gain (ADG), average daily feed intake (ADFI), and the ratio of feed to gain (F/G) were calculated at different stages.

### 2.2. Blood Analysis

At the end of the trial, one pig from each pen that was closest to the average body weight was selected from a pen (replicate) for slaughter. When the pigs approached the target slaughter weight (105 kg to 110 kg), blood samples were collected by venepuncture at 8:30 h after 12 h of fasting. After centrifugation (3500× *g* for 15 min at 4 °C), serum samples were collected and stored at −20 °C until analyzed [20]. The concentrations of glucose (GLU), cholesterol (CHO), triglyceride (TG), low-density lipoprotein (LDL), and high-density lipoprotein (HDL) were determined using available commercial kits according to Nanjing Jiancheng Bioengineering Institute (Nanjing, China). Insulin and insulin-like growth factor-1 (IGF-1) levels were determined using the enzyme-linked immunosorbent assay (ELISA) kits that were purchased from Jiangsu Jingmei Biological Technology Co., Ltd. (Yancheng, Jiangsu, China), and the specific operations were as per the instructions of the kits. Serum antioxidant parameters, including malondialdehyde (MDA), catalase (CAT), total antioxidant capacity (T-AOC), total superoxide dismutase (SOD), and glutathione peroxidase (GSH-Px) were determined using the commercial kits purchased from Nanjing Jiancheng Bioengineering Institute (Nanjing, China) according to the instructions of the kits, and UV-VIS Spectrophotometer (UV1100, MAPADA, Shanghai, China).

### 2.3. Carcass Traits

At the end of the trial, the pigs from which blood samples were taken in wire mesh cages, were transported to the slaughterhouse (Teaching experimental base of Sichuan Agricultural University) in 10-min journeys. Pigs were slaughtered by exsanguination after electrical stunning exsanguinated, dehaired, eviscerated, and split down the midline according to standard commercial procedures. Hot carcass weight was individually recorded and used to calculate the dressing percentage [21]. The other carcass traits’ measurements (obtained from the left side of the carcass) included loin-eye area, average (average of first- and last-rib and last-lumbar fat thickness) backfat thicknesses, carcass length. The loin-eye area (LEA) was measured by tracing the outline of longissimus dorsi muscle (LM) onto transparent paper and then measuring the area using a planimeter, and the average of the 3 measurements was reported as the LEA for each carcass. Within 10 min of slaughter, about 150 g of LM and liver sample from the right side of the carcass were taken anterior to the 10th rib and frozen at 20 °C until lyophilization for muscle chemical analysis. Meanwhile, an approximately 1-cm-thick LM and liver samples were taken posterior to the 10th rib and were frozen in liquid nitrogen until used for RNA extraction.

### 2.4. Meat Quality

Approximately 45 min and 24 h after slaughter, a longissimus section (from 8th to 10th ribs) was removed. Meat color (L*, a*, b*) was measured with a Minolta chromameter (CR300, Minolta Camera CO., Osaka, Japan). Muscle pH_45min_ and pH_24h_ were measured by using a pH meter. The pH meter was calibrated according to a set of calibration standards, including pH 4.00/6.86/9.18. Each chop was measured three times in different areas, and the average value was obtained. A 2.50 cm section of the 10th-rib chop was then removed, and drip loss was determined by a suspension method [22], that is, the LM was removed, trimmed of fat, weighed, suspended by a hook in a Whirl-Par bag, sealed, and stored at 4 °C, and after 24 h, the LM was weighed, and the drip loss was calculated. Drip loss was calculated as follows: %drip loss = [(initial weight − final weight)/initial weight] × 100. The pork tenderness was measured as below. Briefly, LM chops were thawed for 16 h at 4 °C and then cooked to an internal temperature of 70 °C in 80 °C thermostatic water-bath. The internal temperature was monitored during cooking with a handheld thermometer. Chops were allowed to cool to room temperature. Five cooked chops were cut to 1 × 1 × 3 cm 2 parallel to the muscle fiber orientation to measure the tenderness using a Tensipresser (TTP-50BXII, Taketomo Electric Corp., Tokyo, Japan). Samples (5 g) in triplicate were removed from each pig, and cooking weight was determined. The LM meat has cooked for 30 min in the top of a double boiler (sealed) and then cooled 15 min to room temperature. Cooking loss was calculated as a percentage of lost weight based on the pre-cooking weight of a sample. Lipid concentrations of LM and liver were analyzed by a Soxhlet extraction procedure using petroleum ether as an extraction agent. Glycogen concentrations of LM and liver were determined in accordance with the kit purchased from Nanjing Jiangcheng Company [23]. Briefly, after taking fresh liver or muscle samples and rinsing with physiological saline, the filter paper was blotted dry and weighed (sample weight ≤ 100 mg). Hydrolysis. The glycogen hydrolysate was further prepared into a glycogen detection solution: distilled water, 0.01 mg/mL standard solution, glycogen detection solution and color developing solution, and then boiled in boiling water for 5 min. After cooling, the OD (optical density) value of each tube was measured. The alkaline solution, 0.01 mg/mL and the color developing solution are all provided by Nanjing Jiancheng Bioengineering Institute (Nanjing, China).

### 2.5. RNA Extraction and Real-Time RT-PCR

Total RNA was isolated from the longissimus dorsi using TRIzol (TaKaRa, Dalian, China), and then reverse transcription reactions were performed using the Prime Script™ reagent kit (TaKaRa, Dalian, China). There are five steps in the extraction of the RNA: (1) Grinding the sample, put about 1 g of tissue sample into the mortar for grinding. (2) Add 200 μL of chloroform to the tube, fully precipitate the protein. (3) At this time, the liquid is divided into 3 layers. Pipette 400 μL of the supernatant into a new tube, add 400 μL of isopropanol and mix for 10 min. (4) After centrifugation, the RNA has settled. (5) After RNA extraction is completed, considering the ideal absorbance ratio (1.8 ≤ A260/280 ≤ 2.0). The integrity of RNA was checked by electrophoresis on a 1.5% agarose gel. The RNA samples were reverse transcribed into complementary DNA using the PrimeScripte RT reagent kit (Takara) according to the manufacturer’s instructions. The complementary DNA was diluted and used as a PCR template to evaluate gene expression. The primers were synthesized commercially by TaKaRa Biotechnology (Dalian) Co., Ltd. (Dalian, China). Real-time PCR primers were designed by Sangon Biotech (Shanghai, China) to assay three genes related to protein metabolism (*mTOR, MuRF-1, Atrogin1* (for longissimus dorsi)), three genes related to sugar and lipid metabolism (the glucokinase (*GCK*), hormone-sensitive lipase (*HSL*), fatty acid synthase (*FASN*)), four to muscle fiber types (*MyHC I, IIa, IIx, IIb*) and *β-actin* (housekeeper genes), as shown in Table 2. Real-time RT-PCR for the seven target genes and the house-keeping gene were performed using Applied Biosystems (Applied Biosystems, Foster City, CA, USA) Power SYBR Green PCR Master Mix in a Bio-Rad iCycle with minor modifications (Bio-Rad, Hercules, CA, USA). Fluorescein was added at a final concentration of 10 nm as the reference dye. Cycling conditions were as follows: 5 °C for 30 s, followed by 40 cycles at 95 °C for 5 s, 60 °C for 34 s, under melt curve conditions at 95 °C for 15 s, 60 °C for 1 min, and then 95 °C for 15 s (temperature change velocity 0.5 °C/s).

### 2.6. Statistical Analyses

For the statistics of experimental data, such as the ADG and ADFI, all the pigs were used (each group contained 6 replicates (pen), and each replicate contained 3 pigs). After the trial, one pig from each pen was selected for slaughter (*n* = 6). Gene expression data from replicate samples were averaged and analyzed using the 2^−ΔΔCt^ method [24] to measure the difference between the two diets. Growth performance, serum data, carcass traits, and meat quality data were analyzed using the normal distribution procedure of SPSS 22.0 software (SPSS, Chicago, IL, USA). The statistical difference among them was determined by Student’s *t*-tests. Results were presented as the mean values and the standard error of the mean. *p*-value < 0.05 was considered to be significant, whereas a *p*-value between 0.05 and 0.10 was classified as a trend.

## 3. Results

### 3.1. Growth Performance

As shown in Table 3, there were no significant differences in ADG, ADFI, and F:G between the CON and the inulin (INU) group during growing the period (1–32 d). At the early fattening stage (32–70 d), dietary INU supplementation significantly decreased the ratio of F:G (*p* < 0.05). Moreover, INU supplementation also tended to increase the ADG at this stage (*p* = 0.06). During the last fattening stage (70–96 d), INU supplementation significantly increased the ADFI (*p* < 0.05) as well and tended to increase the ADG (*p* = 0.08).

### 3.2. Serum Metabolites, Hormones, and Antioxidant Capacity

Dietary INU supplementation had no significant influences on serum triglycerides (*p* > 0.10), total cholesterol, HDL, and glucose (Table 4). However, the serum concentrations of LDL, insulin, and IGF-1 were significantly higher in the INU group than in the control group (*p* < 0.05). Dietary INU supplementation had no significant influences on antioxidant indices, such as the activities of catalase (CAT), total-superoxide dismutase (T-SOD), and glutathione peroxidase (GSH-Px).

### 3.3. Carcass Traits and Meat Quality

The carcass traits are shown in Table 5. As compared to the control group, the diet with INU supplementation significantly increased the carcass weight (*p* < 0.05) and the dressing percentage (*p* < 0.05). Moreover, INU supplementation tended to increase the loin-eye area (*p* = 0.10). Dietary INU supplementation had no significant influences on meat quality parameters, such as the color, drip loss, and shear force (*p* > 0.05). Moreover, INU supplementation had no significant influences on the contents of crude fat and glycogen in longissimus dorsi muscle and liver (Table 6).

### 3.4. Expression of Genes Related to Metabolism

Dietary INU supplementation significantly elevated the expression level of *MyHC IIb* in the longissimus dorsi (*p* < 0.05). There were no significant differences in the expression levels of other myofiber-related genes, such as the *MyHC I, MyHC IIx*, and *MyHC IIa*, between the two groups (Table 7). Moreover, as shown in Table 7, dietary INU supplementation had no significant influences on glucose and lipid metabolic genes, such as the glucokinase (*GCK*), hormone-sensitive lipase (*HSL*), and fatty acid synthase (*FASN*) in the longissimus dorsi (*p* > 0.05). However, INU significantly upregulated the expression level of rapamycin target protein in mammalian cells (*mTOR*) and decreased the expression level of muscle-specific ubiquitin ligase MuRF-1 (*p* < 0.05).

## 4. Discussion

In recent years, dietary fibers have attracted considerable research interest since they were found to play a critical role in regulating gut health and metabolisms [25]. In the present study, dietary supplementation of 0.5% inulin tended to increase the average daily gain (ADG) and significantly decreased the ratio of F:G at the early fattening stage (32–70 d), suggesting a beneficial role of inulin in the regulation of animal growth. The result is consistent with a previous study that oligosaccharides, such as oligofructose and oligofructose, significantly promoted the growth of pigs in the last three weeks [26]. This may be attributed in part to the difference in the result of the first phase during the animal trial. In this study, we found that serum IGF-1 and insulin concentrations were both significantly elevated after inulin supplementation. The improved ADG at the growth and fattening stage (32–96 d) by inulin is due in part to the elevated IGF-1 and insulin levels in the serum. IGF-1 is an important regulator of growth hormone in promoting growth [27]. Insulin is the main regulator of blood glucose concentration, increasing glucose uptake by muscle and fat, and inhibiting hepatic glucose production [28]. Previous studies revealed that insulin rapidly activates protein synthesis by activating components of the translational machinery, including (eukaryotic initiation factors) eIFs and eEFs (eukaryotic elongation factors) [29]. Insulin has also been reported to increase the cellular content of ribosomes to augment the capacity for protein synthesis [30].

Previous studies have indicated that dietary fiber plays a critical role in regulating carcass quality [31]. For instance, dietary supplementation of pea fiber increased the weight of longissimus dorsi muscle [32]. Moreover, dietary supplementation of sugar beet pulp increased the dressing percentage, loin muscle area, and lean percentage in growing–finishing pigs [33]. In the present study, we found that dietary inulin supplementation significantly increased the carcass weight and the dressing percentage, and tended to increase loin muscle area. The improved carcass traits may also be associated with the elevated IGF-1 and insulin levels in the body after inulin ingestion. A previous study reported that IGF-1 regulates the growth of skeletal muscle and stimulates the terminal differentiation of myoblasts by inducing the expression of cytogenetic genes [34].

In the present study, dietary inulin supplementation had no significant influences on other parameters related to meat quality, such as the color, drip loss, and shear force. However, dietary inulin supplementation significantly elevated the expression level of *MyHC IIb* in the longissimus dorsi. Previous studies indicated that there are four major myofiber types in adult pig skeletal muscles, namely slow-oxidative type I, fast oxide-glycolytic type IIA, and fast glycolytic types IIX and IIB [35], which are encoded by myosin heavy chain (*MyHC*) isoform genes: *I, IIa, IIx,* and *IIb*, respectively [36,37]. The *MyHC IIb* gene encodes a fast-glycolytic type of myofiber, which can generate a large amount of energy in a short time through the creatine phosphate and glycogenolysis pathways, and thus can perform short-term high-intensity contraction movement [38]. In this study, the elevated expression level of *MyHC IIb* suggests that inulin supplementation may promote muscle hypertrophy since there is a positive correlation between the degree of muscular development and the expression level of *MyHC IIb* in pigs [39].

We found that inulin supplementation significantly altered the expression level of several protein metabolic-related genes. The rapamycin target protein in mammalian cells (*mTOR*) was found to play a critical role in the regulation of protein synthesis through its downstream targets S6K1 and 4EBP1 [40]. In this study, the expression level of *mTOR* was significantly elevated by inulin, which may provide a molecular basis behind the elevated growth performance and dressing percentage during the fattening period. Moreover, the expression level of *mTOR* was also consistent with the elevated serum IGF-I and insulin level after inulin supplementation since the IGF-I was reported to promote protein synthesis via stimulating the *mTOR* signaling pathway [41]. While the insulin stimulates protein synthesis and cell growth by activating both the protein kinases Akt (also known as protein kinase B, PKB) and mammalian target of rapamycin (*mTOR*) [42]. Moreover, we found that the muscle-specific ubiquitin ligase *MuRF-1* gene was significantly downregulated by inulin. *MuRF-1* is a ubiquitin ligase E3 associated with skeletal muscle atrophy [43]. This ring finger protein was initially found in association with the myofibril [44] and thus may play an important role in the breakdown of myofibrillar proteins. Our result is also consistent with a previous study showing that IGF-I stimulates muscle growth by suppressing protein breakdown and expression of atrophy-related ubiquitin ligases, *atrogin-1*, and *MuRF1* [45].

## 5. Conclusions

In summary, dietary inulin supplementation improves average daily body weight gain (ADG) in 32–96 d, and carcass yield and dressing percentage in growing–finishing pigs, which is associated with elevated serum IGF-I and insulin levels. Moreover, dietary inulin supplementation had no significant influences on meat quality, except for the expression of the *MyHC IIb* gene, which could alter the myofibers type distribution. The results show the beneficial effect of inulin supplementation on the growth performance and carcass traits in growing–finishing pigs, which will also facilitate the application of inulin in swine production.

## Figures and Tables

**Table 1 animals-09-00840-t001:** Composition and nutrients levels of the basal diet (air-dry basis, %).

Ingredients, %	1–32 d	32–70 d	70–96 d
Corn	76.55	80.30	84.00
Soybean meal	16.71	15.43	10.87
Wheat bran			1.00
Fish meal	2.70		
Soybean oil	1.40	1.40	1.50
Limestone	0.73	0.70	0.63
CaHPO_4_	0.47	0.66	0.53
NaCl	0.30	0.35	0.35
L-Lysine HCl	0.49	0.51	0.48
DL-Methionine	0.08	0.07	0.07
L-Threonine	0.15	0.16	0.15
L-Tryptophan	0.04	0.04	0.04
Choline chloride	0.15	0.15	0.15
Vitamin premix ^1^	0.03	0.03	0.03
Mineral premix ^2^	0.20	0.20	0.20
Total	100.00	100.00	100.00
Nutrients levels ^3^			
DE (Mcal/kg)	3.36	3.36	3.37
CP (%)	15.69	13.75	12.13
CF (%)	2.22	2.20	2.05
Ca (%)	0.66	0.59	0.52
TP (%)	0.50	0.46	0.42
AP (%)	0.31	0.27	0.2
D-Lys	1.03	0.90	0.78
D-Met	0.30	0.25	0.23
D-Met + Cys	0.49	0.43	0.39
D-Thr	0.62	0.55	0.49

Inulin: DE (Mcal/kg) = 3.94; CP (%) = 0.28; TP (%) = 0.03; AP (%) = 0.01. CP, Crude protein; TP, Total p; AP, available p. ^1^ Vitamin premix provided the following per kilogram of diets: Vitamin A, 9000 IU; Vitamin D3, 3000 IU; Vitamin E, 20 IU; Vitamin K3, 3.0 mg; Vitamin B1, 1.5 mg; Vitamin B2, 4.0 mg; Vitamin B6, 3.0 mg; Vitamin B12, 0.02 mg; Niacin, 30 mg; Pantothenic, 15 mg; Folic acid, 0.75 mg; Biotin, 0.1 mg. ^2^ Mineral premix provided the following per kg of diets, 1–32 d: Fe (FeSO_4_·H_2_O) 60 mg, Cu(CuSO_4_·5H_2_O) 4 mg, Mn(MnSO_4_·H_2_O) 2 mg, Zn(ZnSO₄·H₂O) 60 mg, I(KI) 0.14 mg, Se(Na_2_SeO_3_) 0.2 mg; 32–70 d: Fe(FeSO_4_·H_2_O) 50 mg, Cu(CuSO_4_·5H_2_O) 3.5 mg, Mn(MnSO_4_·H_2_O) 2 mg, Zn(ZnSO₄·H₂O) 50 mg, I(KI) 0.14 mg, Se(Na_2_SeO_3_) 0.15 mg; 70–96 d: Fe(FeSO_4_·H_2_O) 40 mg, Cu(CuSO_4_·5H_2_O) 3 mg, Mn(MnSO_4_·H_2_O) 2 mg, Zn(ZnSO₄·H₂O) 50 mg, I(KI) 0.14 mg, Se(Na_2_SeO_3_) 0.15 mg. ^3^ Values are calculated.

**Table 2 animals-09-00840-t002:** Primers used for real-time quantitative PCR.

Gene	Accession Number	Primer Sequence (5’–3’)	Size (bp)
*β-actin*	XM_003124280.5	F: TGGAACGGTGAAGGTGACAGC R: GCTTTTGGGAAGGCAGGGACT	177
*mTOR*	XM_003127584.6	F: GCACAAGGACGGATTCCTAC R: CACTTGCGTTGGGACATC	248
*MuRF-1*	NM_001184756.1	F: AACCTGGAGAAGCAGCTGAT R: TAGGGATTTGCAGCCTGGAA	128
*Atrogin1*	NM_001044588.1	F: TGGACTTCTCGACTGCCATT R: GCTATCAGTTCCAACAGCCG	70
*HSL*	NM_214315.3	F: CACAAGGGCTGCTTCTACGG R: AAGCGGCCACTGGTGAAGAG	167
*FASN*	NM_001099930.1	F: CTACGAGGCCATTGTGGACG R: AGCCTATCATGCTGTAGCCC	146
*GCK*	XM_013985832.2	F: ATCAAACGGAGAGGGGACTT R: ACAATCATGCCAACCTCACA	113
*MyHC I*	NM_213855.1	F: GTTTGCCAACTATGCTGGGG R: TGTGCAGAGCTGACACAGTC	95
*MyHC IIa*	NM214136.1	F: CTCTGAGTTCAGCAGCCATGA R: GATGTCTTGGCATCAAAGGGC	127
*MyHC IIb*	NM_001123141.1	F: GAGGTACATCTAGTGCCCTGC R: GCAGCCTCCCCAAAAATAGC	83
*MyHC IIx*	NM_001104951.2	F: TTGACTGGGCTGCCATCAAT R: GCCTCAATGCGCTCCTTTTC	111

*mTOR* = rapamycin target protein in mammalian cell; *MuRF-1* = Muscle RING finger 1; *Atrogin1* = Atrophy Gene 1; *HSL*
*=* hormone-sensitive lipase; *FASN* = fatty acid synthase; *GCK* = glucokinase; *MyHC I* = heavy myosin-chain I; *MyHC IIa* = heavy myosin-chain IIa; *MyHC IIb* = heavy myosin-chain IIb; *MyHC IIx* = heavy myosin-chain IIx; F = forward; R = reverse.

**Table 3 animals-09-00840-t003:** Effect of dietary supplementation with inulin on growth performance in growing–finishing pigs ^†^.

Items	CON	INU	*p*-Value
1–32 d			
Initial weight, kg	22.28 ± 0.29	22.26 ± 0.27	0.967
Final weight, kg	45.22 ± 0.95	44.30 ± 0.51	0.414
ADG, g/d	716.88 ± 32.32	688.59 ± 16.76	0.455
ADFI, g/d	1434.98 ± 25.49	1434.64 ± 49.49	0.995
F/G	2.00 ± 0.06	2.08 ± 0.04	0.385
32–70 d			
Initial weight, kg	45.22 ± 0.95	44.30 ± 0.51	0.414
Final weight, kg	81.62 ± 1.82	84.16 ± 1.58	0.316
ADG, g/d	933.29 ± 24.25	1022.05 ± 34.59	0.062
ADFI, g/d	2610 ± 60.36	2650.30 ± 97.15	0.732
F/G	2.80 ± 0.07 ^b^	2.59 ± 0.06 ^a^	0.045
70–96 d			
Initial weight, kg	81.62 ± 1.82	84.16 ± 1.58	0.316
Final weight, kg	108.05 ± 2.31	112.36 ± 1.79	0.171
ADG, g/d	1057.4 ± 23.20	1128.33 ± 28.31	0.081
ADFI, g/d	3086.52 ± 67.97 ^b^	3336.51 ± 76.50 ^a^	0.035
F/G	2.92 ± 0.07	2.96 ± 0.03	0.647
1–96 d			
Initial weight, kg	22.28 ± 0.29	22.26 ± 22.26	0.967
Final weight, kg	108.05 ± 2.31	112.36 ± 1.79	0.171
ADG, g/d	893.47 ± 25.60	938.58 ± 18.42	0.183
ADFI, g/d	2342.42 ± 42.16	2436.11 ± 49.31	0.179
F/G	2.63 ± 0.06	2.59 ± 0.03	0.649

^†^ Values are expressed as the mean of six replicates (pen) in each group. ^a,b^ Mean values within a row with unlike superscript letters were significantly different (*p* < 0.05).

**Table 4 animals-09-00840-t004:** Effect of dietary inulin supplementation on serum metabolites, hormones, and antioxidant capacity in growing–finishing pigs ^†^.

Items	CON	INU	*p*-Value
Triglycerides, mmol/L	0.42 ± 0.02	0.49 ± 0.04	0.161
Total cholesterol, mmol/L	3.74 ± 0.22	3.65 ± 0.11	0.705
HDL, mmol/L	5.30 ± 0.20	5.61 ± 0.38	0.503
LDL, mmol/L	1.25 ± 0.03 ^b^	1.51 ± 0.09 ^a^	0.019
Insulin, mIU/L	48.61 ± 1.34 ^b^	53.63 ± 1.67 ^a^	0.042
Glucose, mmol/L	4.15 ± 0.18	3.91 ± 0.32	0.531
IGF-1, ug/L	2.19 ± 0.06 ^b^	2.32 ± 0.11 ^a^	0.025
GSH-Px, U/mL	1613.25 ± 106.16	1726.86 ± 107.18	0.468
MDA, nmol/mL	2.00 ± 0.09	1.93 ± 0.09	0.567
CAT, U/mL	6.77 ± 0.49	7.46 ± 0.89	0.518
T-AOC, U/mL	4.99 ± 0.56	3.93 ±0.42	0.074
SOD, U/mL	71.77 ± 4.86	76.63 ± 2.60	0.403

^†^ Values are expressed as the mean of six pigs in each group. ^a,b^ Mean values within a row with unlike superscript letters were significantly different (*p* < 0.05).

**Table 5 animals-09-00840-t005:** Effect of dietary inulin supplementation on carcass traits and meat quality in growing–finishing pigs ^†^.

Items	CON	INU	*p*-Value
Final body weight, kg	108.05 ± 2.31	112.36 ± 1.79	0.171
Carcass weight, kg	75.13 ± 2.50 ^a^	81.13 ± 1.44 ^b^	0.025
Dressing percentage, %	69.52 ± 0.95 ^a^	72.20 ± 0.24 ^b^	0.040
Carcass length, cm	101.80 ± 0.96	99.40 ± 1.31	0.111
Backfat depth, mm	26.16 ± 2.57	29.76 ± 2.90	0.309
LEA, cm^2^	51.69 ± 1.52	60.52 ± 4.29	0.097
pH_45min_	6.66 ± 0.17	6.48 ±0.07	0.376
pH_24h_	6.07 ± 0.02	6.06 ± 0.01	0.472
L*_45 min_	42.75 ± 0.45	41.40 ± 0.60	0.114
L*_24 h_	50.93 ± 1.29	49.78 ± 0.71	0.457
a*_45 min_	5.54 ± 0.34	4.71 ± 0.20	0.074
a*_24 h_	8.91 ± 0.64	8.16 ± 0.37	0.338
b*_45 min_	2.43 ± 025	2.22 ± 0.27	0.592
b*_24 h_	7.70 ± 0.76	9.77 ± 0.48	0.793
Drip loss, %	2.18 ± 0.32	2.05 ± 0.12	0.453
Cook loss, %	35.38 ± 1.21	35.02 ± 0.40	0.788
Shear force, kg	3.47 ± 0.17	3.12 ± 0.16	0.342

^†^ Values are expressed as the mean of six pigs in each group. ^a,b^ Mean values within a row with unlike superscript letters were significantly different (*p* < 0.05).

**Table 6 animals-09-00840-t006:** Effect of dietary inulin supplementation on the contents of crude fat, glycogen in the longissimus dorsi muscle and liver, and in growing–finishing pigs ^†^.

Items	CON	INU	*p*-Value
Liver glycogen content, %	7.74 ± 0.40	7.65 ± 0.55	0.893
Muscle glycogen content, %	1.33 ± 0.10	1.53 ± 0.08	0.185
Liver crude fat content, %	5.37 ± 0.22	5.66 ± 0.31	0.486
IMF content, %	3.14 ± 0.16	3.38 ± 0.17	0.337

^†^ Values are expressed as the mean of six pigs in each group.

**Table 7 animals-09-00840-t007:** Effect of dietary inulin supplementation on the expression of myosin heavy chain (MyHC) isoform genes and metabolic genes in longissimus dorsi ^†^.

Items	CON	INU	*p*-Value
MyHC I	1.00 ± 0.17	0.78 ± 0.16	0.583
MyHC IIx	1.00 ± 0.23	1.06 ± 0.09	0.943
MyHC IIa	1.00 ± 0.21	0.98 ± 0.09	0.934
MyHC IIb	1.00 ± 0.12	1.65 ± 0.17	0.029
GCK	1.00 ± 0.15	0.92 ± 0.22	0.728
HSL	1.00 ± 0.09	1.53 ± 0.15	0.560
FASN	1.00 ± 0.23	0.90 ± 0.11	0.358
mTOR	1.00 ± 0.13	1.78 ± 0.18	0.024
Atrogin1	1.00 ± 0.19	0.73 ± 0.08	0.323
MuRF-1	1.00 ± 0.19	0.52 ± 0.04	0.001

^†^ Values are expressed as the mean of six pigs in each group.

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
