# Peer review of "Effect of Dietary Inulin Supplementation on Growth Performance, Carcass Traits, and Meat Quality in Growing–Finishing Pigs"

_animals, 2019, doi:10.3390/ani9100840_

Round 1

Reviewer 1 Report

Please refer to annotated manuscript for comments.

Author Response

Reviewer 1

1.line11-12 This sentence implies that the meat quality currently does not meeting consumer expectation. Sentence needs to be structured better.

Re: Thanks for your suggestion, we have re-written this sentence.

line14/line30 but inulin had no effect on meat quality

Re: Although, inulin supplementation had no significant influences on meat quality parameters such as the drip loss and color. However, it significantly altered the distribution of muscle fibers (myofiber types are closely associated with meat quality) and increased the lean rate. I accept your suggestion and revised the description (i.e line30, we changed the “meat quality” to “carcass trait”).

line32 Need to standardise terms through out this paper - either meat quality or pork quality.

Re: We have standardized the decription as the “meat quality” throughout the text.

line 35-64 Introduction is very broad and does not really articulate the issue/aspects of meat quality. Hypothesis is very general. This section needs a we-write.

Re: We agree with the reviewer and we have re-written this section.

line 43 which species?

Re: We have added it in the revised text (human, mice and pigs).

line62 Need to expand this sentence - what similar role and what aspect of meat quality.

Re: We have revised this sentence in the text (We have already cited some previous studies in the introduction section. For clearance, this sentence has been removed).

line76 group?

Re: Each group contains six replicates with 3 pigs per replicate (the 3 pigs were housed in a pen).

line126 reference

Re: We added the reference in the text (see line186).

9.line162 Given the sample size, and the experimental design, was the pen or the pig the experimental unit in analysing the growth and pork quality data?

Re: For the statistics of growth performance (i.e. ADG) all the pigs were used [each group contains 6 replicates (pen), and each replicate contains 3 pigs]. However, only one pig from each pen was selected for slaughter (n=6). This has been stated in the revised manuscript.

line230 Yes, there were some ADG increases but there was no life performance gains. Authors may be presuming too much by stating their data was consistent with the referenced literature.

Re: In this study, inulin tended to improve the ADG (P=0.062) at the early fattening stage (1-32d). As shown in table3, we also recorded the initial and final body weight (from 1d to 96d). However, no significant difference was observed.

line252 why is this astonishing? If you look at the meat quality parameter data, they appear to have large SEM. Suggest that the sample size was not sufficient.

Re: Actually, for most parameters, the SEM is acceptable. Moreover, six replicates are sufficient to meet the minimal requirement of statistics for mechanism study. In this study, inulin increased the dressing percentage and altered muscle fiber distribution, but had no significant influences on some meat quality parameters such as the color, drip loss and shear force. Our result is consistent with previous reports by using different dietary fibers [1,2]. To aviod confusing, we have revised this sentence in the text.

【1】. Kass, M. L., Soest, P. J. V., Pond, W. G., Lewis, B., & Mcdowell, R. E. Utilization of dietary fiber from alfalfa by growing swine. i. apparent digestibility of diet components in specific segments of the gastrointestinal tract. Journal of Animal Science. 1980, 50, 175-191.

【2】.Kjosetal, N. P., Overland, M., Matre, T. Pig feed from sugar beet pulp[J]. Feed Mix. 1999, 7(4): 22- 24.

——We have corrected all necessary language and grammar mistakes based on reviewer’s suggestions.

Reviewer 2 Report

The objective of this paper was to explore the effects of dietary inulin supplementation on growth performance, blood metabolic parameters, carcass traits, and meat quality in growing-finishing pigs. 32 pigs (two diets by 6 replicates of 3 pigs per pens). The authors conclude that inulin at 0.5% is beneficial for pig growth performance, but also indicate potential mechanisms for the effects of inulin on pork quality and metabolism of growing-finishing pigs is via increases IGF-1, mTOR and MyHC- IIb. In short, this paper is a easy read, and designed well. However, the authors need to provide more details and especially diet analysis. Also, the authors need to discuss the biological significance of their data as although some parameters are significantly different, biologically they may not mean much.

Specific comments:

Line 38: … produced inulin is extracted …. Line 44: Gut health is not SCFA production. Please define gut health better. Line 47: In addition to SCFAs/butyrates beneficial effects …. Line 51-52: Could transition better between paragraphs. Line 56: This paper looked at dietary starch type not fiber. Please cite correct information. Line 70: Herd not flock. Line 80: Feed distribution of the pigs were determined daily throughout the trial on a pen basis? What does this mean? How much in each of the 3 feedings was given? How was refusals determined? Line 73: Add that the diet were fed in 3 phases. Line 162: Please state if pen was the experimental unit. Line 94: randomly selected? Line 97: EDTA or heparin plasma? Line 111: Sichuan Agricultural University Line 113: When was the Longissimus dorsi muscle and liver samples collected in relation … Line 114-119: How was all these carcass measures taken? Details missing. Line 140: How was glycogen concentrations measured? Line 142-145: Details of RNA isolations? How much tissue? Quality? cDNA synthesis? Line 150: and throughout, capital T for Table Line 156: Housekeeper genes? I assume B-actin. Please clarify. Line 175: Mention ADFI here as well. Line 282: IGF-1 0.13 ug/L difference is biological important? Line 281: improved carcass yield and dressing percentage, but did not alter carcass quality measures. Line 28-31: Not quite accurate conclusion. Mechanism is not shown. Discussion: The authors need to discuss inclusion rates when comparing studies. Discussion: IGF-1 data is over interpreted. Please address. Line 245: Reference needed. Line 251: How? Line 252: Why astonished? Table 3: Express ADG and ADFI on kg/d basis. Table 1: Please should both the Control and Inulin diet formulations. Was it added in place of corn? Please report the formulated fiber information. The authors should also report the analyzed fiber of inulin content of the diets. What is CP, CA, TP, AP? Spell out. Figure 1, 2, 3. Y-axis units? Change axis title. Axis titles the same in figure 2 and 3? Should combine all figures into a table to save space.

Author Response

Reviewer 2

The objective of this paper was to explore the effects of dietary inulin supplementation on growth performance, blood metabolic parameters, carcass traits, and meat quality in growing-finishing pigs. 32 pigs (two diets by 6 replicates of 3 pigs per pens). The authors conclude that inulin at 0.5% is beneficial for pig growth performance, but also indicate potential mechanisms for the effects of inulin on pork quality and metabolism of growing-finishing pigs is via increases IGF-1, mTOR and MyHC- IIb. In short, this paper is a easy read, and designed well. However, the authors need to provide more details and especially diet analysis. Also, the authors need to discuss the biological significance of their data as although some parameters are significantly different, biologically they may not mean much.

Re: Thanks for your suggestions. We have provided more experimental details in the revised manuscript (see the materials and methods section). Moreover, the discussion part has been revised and biological significance of the data has been added.

Line 38: … produced inulin is extracted

Re: It has been revised in the text.

Line 44: Gut health is not SCFA production. Please define gut health better.

Re: We are sorry for the confused sentences. These sentences have been revised in the text.

Line 47: In addition to SCFAs/butyrates beneficial effects ….

Re: It has been revised in the text.

Line 51-52: Could transition better between paragraphs.

Re: It has been revised according to another reviewer’s suggestion.

Line 56: This paper looked at dietary starch type not fiber. Please cite correct information.

Re: the correct reference has been provided in the revised manuscript.

Line 70: Herd not flock.

Re: It has been revised in the text.。

Line 80: Feed distribution of the pigs were determined daily throughout the trial on a pen basis? What does this mean? How much in each of the 3 feedings was given? How was refusals determined?

Re: Because 3 pigs were housed in a pen (replicate), the feed intake can not be recorded for an individual pig and it has been recorded on a pen basis. Actually, the pigs were fed ad libitum. The pigs were fed three times per day to make sure that the fresh feed was available for pigs. For each feeding time, pigs were fed according to the conditions of each pen (the feeding was stopped when all pigs stopped eating).

Line 73: Add that the diet were fed in 3 phases.

Re: This has been added in the revised manuscript.

Line 162: Please state if pen was the experimental unit.

Re: For the statistics of growth performance (i.e. ADG) all the pigs were used [each group contains 6 replicates (pen), and each replicate contains 3 pigs]. However, only one pigs from each pen was selected for slaughter (n=6). This has been stated in the revised manuscript.

Line 94: randomly selected?

Re: One pig that is closest to the average weight is selected from a pen (replicate) for slaughter. We have added this in the text.

11,Line 97: EDTA or heparin plasma?

Re: Actually, we did not use anticoagulant for blood collection. Because we only isolated the serum sample. This has been stated in the revised manuscript.

Line 111: Sichuan Agricultural University

Re: It has been revised in the text.

13 Line 113: When was the Longissimus dorsi muscle and liver samples collected in relation …

Re: The dorsi muscle and liver were collected within ten minutes after slaughter.

Line 114-119: How was all these carcass measures taken? Details missing.

Re: The details about the carcass measurements have been added in the revised manuscript.

Line 140: How was glycogen concentrations measured?

Re: The glycogen concentration was determined by a commercial kit purchased from Nanjing Jiancheng Biological Company. The experimental procedures were added in the revised manuscript.

17.Line 142-145: Details of RNA isolations? How much tissue? Quality? cDNA synthesis?

Re: The details of the experimental procedures were added in the revised manuscript.

Line 150: and throughout, capital T for Table

Re: It has been revised in the text.

Line 156: Housekeeper genes? I assume B-actin. Please clarify.

Re: β-actin function as a housekeeping gene. This has been clarified in the text.

Line 175: Mention ADFI here as well.

Re: It has been revised in the text.

21 Line 282: IGF-1 0.13 ug/L difference is biological important?

Re: The result from statistics showed that there’s significant difference in serum IGF-1 concentration between the control and inulin group (P<0.05). As we discussed in the manuscript, the IGF-1 is an important regulator of the growth of skeletal muscles and can stimulate terminal differentiation of myoblasts by inducing the expression of cytogenetic genes. The difference in the serum IGF-1 concentration may contribute in part to the elevated growth performance. We revised the description for this sentence in the text.

Line 281: improved carcass yield and dressing percentage, but did not alter carcass quality measures.

Re: This sentence has been revised in the text.

23.. Line 28-31: Not quite accurate conclusion. Mechanism is not shown. Discussion: The authors need to discuss inclusion rates when comparing studies. Discussion: IGF-1 data is over interpreted. Please address.

Re: We agree with the reviewer and have redefined the conclusions. The problem of IGF-1 over-interpretation has also been revised in the article.

Line 245: Reference needed.

Re: The references have been added in the revised manuscript.

25 Line 251: How?

Re: As discussed in the text, IGF-1 is an important regulator of growth hormone in promoting growth [25]. It has been reported to regulate the growth of skeletal muscle and stimulates the terminal differentiation of myoblasts by inducing the expression of cytogenetic genes [26]. Insulin is the main regulator of blood glucose concentration, increasing glucose uptake by muscle and fat and inhibiting hepatic glucose production [27]. Previous studies revealed that insulin rapidly activates protein synthesis by activating components of the translational machinery including (eukaryotic initiation factors) eIFs and eEFs (eukaryotic elongation factors) [28]. Insulin also been reported that increases the cellular content of ribosomes to augment the capacity for protein synthesis [29]. Therefore, the improved growth or carcass traits may be due in part to the elevated serum concentrations of IGF-1 and insulin. These references have been added to this sentence.

Line 252: Why astonished? Table 3: Express ADG and ADFI on kg/d basis.

Table 1: Please should both the Control and Inulin diet formulations. Was it added in place of corn? Please report the formulated fiber information. The authors should also report the analyzed fiber of inulin content of the diets. What is CP, CA, TP, AP? Spell out. Figure 1, 2, 3. Y-axis units? Change axis title. Axis titles the same in figure 2 and 3? Should combine all figures into a table to save space.

Re: This sentence has been revised in the text. The words I used here are not appropriate enough. I have modified the words in the text.

Table 1: It is not necessary to show the inulin diet formulation since it has been prepared on the basis of the control diet. The inulin was added to the basal diet at a dosage of 0.5% (99.5% basal diet + 0.5% inulin). The crude fiber has been provided in the Table1. The full name of CP, CA, TP, AP were given in the revised manuscript. All the figures have been turned into table based on the reviewer’s suggestion.

Reviewer 3 Report

General comment

Improving of pork production efficiency is an important goal of animal science and production worldwide. One of the most important issue is the influence of nutritional factors on health, welfare and production parameters including fattening and slaughter performance. The paper being reviewed is the part of mentioned scientific trend that is why seems to be in compliance with the scope of the journal. The manuscript is written proper language and needs only minor language revision. Unfortunately, many imprecision and unclear information were found, especially in M&M, what makes it difficult to assess real value of research and paper.

Specific comments

Introduction

Introduction is relatively short but well written and informative section. However, reading next sections one can have feeling that it is not fully compatible with M&M and results. Introduction should lead the reader straight to the aim of the study, and justify the research and experimental design. At least 2 parts of the research (blood analyses and gene expressions) are not mentioned in Introduction even with one sentence. This should be corrected and completed.   

M&M

Undoubtedly this section is the worst in whole manuscript. The most of information is unclear, many important data is missed or described imprecisely. This make all the manuscript difficult to assess in terms of data reliability and correctness of conclusions. Whole M&M section needs deep, substantial improvement.

Line 69: the sequence of crossing breeds is important for final effect, and should be described more precisely. I can guess that parents were F1 sows Landrace x Yorkshire mated with Duroc boar, but this should be written, not left for speculation. It is also important to clarify origin of breeds. Completely different results will be expected if breeds are from Denmark, and different if they are from e.g. eastern Europe or United States.

Line 74: nothing is known about the source of inulin. It was purchased from industry, it is clear, but it would be important to know what initially was the source (cereals, vegetables, chicory etc.).

Line 75: This is impossible that there was no discrepancy in basal ingredients between control and experimental diets. 5g/kg is not much, but this 5g of something had to be withdrawn from experimental diet and this should be clarified.

Line 81: days of control weighing should be precisely defined. Also methodological assumption to finish the experiment should be defined (was it body mass or time of fattening?). If body mass, the time of fattening should be mentioned as another data in tables to compare.

Line 94: Why only 1 pig per pen was slaughtered? In whole experiment 36 pigs were utilized (18 per group). It means that only 12 pigs were slaughtered and analyzed in further stages of experiment (6 per group). The most of analyses being performed are not excessively expensive, and it is difficult to understand why relatively small initial number of animals (nutritional experiments usually utilize at least 25 animals per group), was again decreased in the most important stages of analyses. This information must be clarified and justified. Maybe I have understand wrong.

Line 121: information unclear. Did Authors collected MLD samples twice? It is difficult to understand why. It is not necessary to collect sample for pH after 45 min. analysis. Analysis can be made on carcass. The rest of quality analyses do not make a sense, because meat properties are important after 24 hours of cooling.

Line 125: what does it mean “different areas”. Can Authors define them more precisely?

line 163-167: this subsection is described very general. It is not so important which software was used. Much more important and needed information are: 1. Were data parametric or nonparametric?; 2. Which statistical tests were used to evaluate distribution of data?; 3. Which tests were used to analyse differences between groups? All this information is missed, and must be completed.

Results

Results are stated in 3 tables and 3 figures. In my opinion figures are not enough precise, and the all the data should be presented in tables. In such a simple experimental design I would also calculate SEM separately for data inside groups. Sometimes such an analyses may show interesting, additional relations (e.g. if some parameters in one group are more dispersed than in the second one it is also important information). P value in tables should be presented as 3 numbers after coma to clarify significance (e.g. in table 3, 50-80kg F/G P-value is 0.05, and it is not clear if it is tendency or significance – 0.049 would be significance, but 0.051 only tendency, and both of them in 2 points after coma have the same value 0.05). And last but not least, in every table n value for each group should be presented, because of imprecise description in M&M. The reader must know the number of animals in particular analyses to assess reliability and value of data.

Discussion

The most of this section seem to be continued Introduction rather than actual discussion. Most of information from the 1st paragraph should be moved to Introduction. The majority of discussion is focused on gene expressions, which in my opinion was only additional analyses of minor importance (they were even omitted in the title). Certainly, I know, that such analyses are fashionable, but the value of information seem to be low, especially in such a small number of analyzed animals. The most important information on fattening performance, carcass traits, and meat quality are only rewritten from results without any trial to explain or interpret. E.g. ADG in the first period was better in control group (not significantly, but it is very difficult to show significance when n=6, however, undoubtedly this difference was relevant for practice). Then the situation were opposite. Such information should be analyzed with the trial to interpret. Why inulin start to improve performance after 50 kg. In my opinion whole the discussion must be completely rewritten to show more critical look for the data, and some trials to interpret them.

Conclusion

Only the first sentence and the first part of the second one seem to be real conclusion. The rest of this section seems to be rather speculation than conclusion and should be moved to discussion (which is the only place for speculation in scientific paper).

To summarize, the manuscript needs substantial improvement, and must be completed with many important data and information. Without this improvement, it is very difficult to assess the reliability of presented data. The most important seem to be clarification of the numbers of animals in consecutive analyses. Taking into account the character of data and experimental design, the number of 6 animals per group seem to be not enough for the paper. I suggest major revision, but this suggestion is conditional. If the number of animals is 6 per group, I would suggest to the Authors to find the other, lower ranked journal, and rewrite the manuscript as a short communication.

Author Response

Reviewer 3

General comment

Improving of pork production efficiency is an important goal of animal science and production worldwide. One of the most important issue is the influence of nutritional factors on health, welfare and production parameters including fattening and slaughter performance. The paper being reviewed is the part of mentioned scientific trend that is why seems to be in compliance with the scope of the journal. The manuscript is written proper language and needs only minor language revision. Unfortunately, many imprecision and unclear information were found, especially in M&M, what makes it difficult to assess real value of research and paper.

Re: We have made necessary revision based on the reviewer’s comments. Especially in M&M section, all the necessary information has been provided in the revised manuscript.

Specific comments

Introduction

Introduction is relatively short but well written and informative section. However, reading next sections one can have feeling that it is not fully compatible with M&M and results. Introduction should lead the reader straight to the aim of the study, and justify the research and experimental design. At least 2 parts of the research (blood analyses and gene expressions) are not mentioned in Introduction even with one sentence. This should be corrected and completed.

Re: Thanks for your suggestion, we have added this missing part to the article.

M&M

Undoubtedly this section is the worst in whole manuscript. The most of information is unclear, many important data is missed or described imprecisely. This make all the manuscript difficult to assess in terms of data reliability and correctness of conclusions. Whole M&M section needs deep, substantial improvement.

Re: Thanks for your suggestion. This section has been revised and all necessary information has been added to the text.

Line 69: the sequence of crossing breeds is important for final effect, and should be described more precisely. I can guess that parents were F1 sows Landrace x Yorkshire mated with Duroc boar, but this should be written, not left for speculation. It is also important to clarify origin of breeds. Completely different results will be expected if breeds are from Denmark, and different if they are from e.g. eastern Europe or United States.

Re: Duroc is terminal boar, Landrace is male parent, Yorkshire is female parent; Duroc is from the United States, Landrace is from Denmark, and Yorkshire is from North England. We have added these information to the article.

Line 74:nothing is known about the source of inulin. It was purchased from industry, it is clear, but it would be important to know what initially was the source (cereals, vegetables, chicory etc.).

Re: The inulin used in this experiment was purchased from Beijing Zhongtaihe technology Co., Ltd. and extracted from chicory as raw material. I have already added it in the article. We have added this missing part to the article.

Line 75: This is impossible that there was no discrepancy in basal ingredients between control and experimental diets. 5g/kg is not much, but this 5g of something had to be withdrawn from experimental diet and this should be clarified.

Re: The inulin was added to the basal diet at a dosage of 5g/kg (99.5% basal diet + 0.5% inulin). This has been clarified in the revised manuscript.

Line 81: days of control weighing should be precisely defined. Also methodological assumption to finish the experiment should be defined (was it body mass or time of fattening?). If body mass, the time of fattening should be mentioned as another data in tables to compare.

Re: The test period is 96 days, and the method for determining the completion of the test is the number of feeding days. This has been added in the Table 1.

Line 94: Why only 1 pig per pen was slaughtered? In whole experiment 36 pigs were utilized (18 per group). It means that only 12 pigs were slaughtered and analyzed in further stages of experiment (6 per group). The most of analyses being performed are not excessively expensive, and it is difficult to understand why relatively small initial number of animals (nutritional experiments usually utilize at least 25 animals per group), was again decreased in the most important stages of analyses. This information must be clarified and justified. Maybe I have understand wrong.

Re: For the statistics of growth performance (i.e. ADG) all the pigs were used [each group contains 6 replicates (pen), and each replicate contains 3 pigs]. However, only one pig (closer to average body weight) from each pen was selected for slaughter (n=6). This has been stated in the revised manuscript. For most biological or mechanism studies, six replicates are sufficient to meet the minimal requirement for statistics.

Line 121: information unclear. Did Authors collected MLD samples twice? It is difficult to understand why. It is not necessary to collect sample for pH after 45 min. analysis. Analysis can be made on carcass. The rest of quality analyses do not make a sense, because meat properties are important after 24 hours of cooling.

Re: All the procedures were based on previous study or standard methods. The LM samples anterior to the 13th rib from the left side carcass were used in the following order: 1) 3.0-cm-thick chop used for objective color measurement (L*, a*, and b*); 2) 3.0-cm-thick chop used for pH measurement; 3) 4.0-cm-thick chop used for drip loss measurement; 4) 4.0-cm-thick chop used for cook loss measurement; and 5) 5.0-cm-thick chop used for shear force measurement. This has been added in the revised manuscript.

Line 125: what does it mean “different areas”. Can Authors define them more precisely?

Re: We are sorry for the confused description. We removed one of the longissimus dorsi muscles and measured three values at three different points of the longissimus dorsi muscle with a pH detector, and then took the average of these three values as the final pH value. The reason for this is to reduce the experimental error. This is a conventional method used for the measurements of pH value [Zybert, A., Protasiuk, E., Antosik, K., Sieczkowska, H., Krzecionieczyporuk, E., & Adamczyk, G., et al. Variations in ph decline measured from 45 min to 48 h postmortem as related to meat quality of (l × y) × h fatteners. Annals of Animal Science. 2014, 14, 461-469.].

line 163-167: this subsection is described very general. It is not so important which software was used. Much more important and needed information are: 1. Were data parametric or nonparametric?; 2. Which statistical tests were used to evaluate distribution of data?; 3. Which tests were used to analyse differences between groups? All this information is missed, and must be completed.

Re: Growth performance, serum data, carcass traits, meat quality data were analyzed using normal distribution procedure of SPSS 22.0 software (SPSS, Chicago, IL, USA). Statistical difference among were determined by Student’s t tests. Results were presented as the mean values and the standard error of the mean. P-value < 0.05 was considered to be significant, whereas a P-value between 0.05 and 0.10 was classified as a trend. This part has been revised in the text.

Results

Results are stated in 3 tables and 3 figures. In my opinion figures are not enough precise, and the all the data should be presented in tables. In such a simple experimental design I would also calculate SEM separately for data inside groups. Sometimes such an analyses may show interesting, additional relations (e.g. if some parameters in one group are more dispersed than in the second one it is also important information). P value in tables should be presented as 3 numbers after coma to clarify significance (e.g. in table 3, 50-80kg F/G P-value is 0.05, and it is not clear if it is tendency or significance – 0.049 would be significance, but 0.051 only tendency, and both of them in 2 points after coma have the same value 0.05). And last but not least, in every table n value for each group should be presented, because of imprecise description in M&M. The reader must know the number of animals in particular analyses to assess reliability and value of data.

Re: Thanks, we agree with the reviewer, and all the data have been presented in tables. The P values have been reserved for three decimal places in all tables. The notes below all the tables have been calculated for six pigs.

Discussion

The most of this section seem to be continued Introduction rather than actual discussion. Most of information from the 1st paragraph should be moved to Introduction. The majority of discussion is focused on gene expressions, which in my opinion was only additional analyses of minor importance (they were even omitted in the title). Certainly, I know, that such analyses are fashionable, but the value of information seem to be low, especially in such a small number of analyzed animals. The most important information on fattening performance, carcass traits, and meat quality are only rewritten from results without any trial to explain or interpret. E.g. ADG in the first period was better in control group (not significantly, but it is very difficult to show significance when n=6, however, undoubtedly this difference was relevant for practice). Then the situation were opposite. Such information should be analyzed with the trial to interpret. Why inulin start to improve performance after 50 kg. In my opinion whole the discussion must be completely rewritten to show more critical look for the data, and some trials to interpret them.

Re: Thanks for your kindly suggestion. The discussion part has been completely revised in the text.

Conclusion

Only the first sentence and the first part of the second one seem to be real conclusion. The rest of this section seems to be rather speculation than conclusion and should be moved to discussion (which is the only place for speculation in scientific paper).

Re: Thanks for your kindly suggestion. The conclusion has been completely revised in the text.

To summarize, the manuscript needs substantial improvement, and must be completed with many important data and information. Without this improvement, it is very difficult to assess the reliability of presented data. The most important seem to be clarification of the numbers of animals in consecutive analyses. Taking into account the character of data and experimental design, the number of 6 animals per group seem to be not enough for the paper. I suggest major revision, but this suggestion is conditional. If the number of animals is 6 per group, I would suggest to the Authors to find the other, lower ranked journal, and rewrite the manuscript as a short communication.

Re: For the statistics of growth performance (i.e. ADG and ADFI) all the pigs were used [each group contains 6 replicates (pen), and each replicate contains 3 pigs]. However, only one pig (closer to average body weight) from each pen was selected for slaughter (n=6). This has been stated in the revised manuscript. For most biological or mechanism studies, six replicates are sufficient to meet the minimal requirement for statistics. This has been completely revised in the text.

Round 2

Reviewer 1 Report

Authors have addressed most of the previous review comments.

Author Response

Thank you for your agreement

Reviewer 2 Report

The authors have done a good job responding to the edits and comments. However, please address the following:

1) Abstract and conclusion - Inulin shown no overall benefit to ADG, but only a tendency in phase 2 and 3. This is not clearly reflected in the conclusions. Please adequately address these.

2) The authors still need to discuss the biological relevance of a 0.13 ug/L difference in IGF-1. I think they are over interpreting their data and this is reflective by the lack of overall growth differences.

3) Conclusion - still not sure on what teh "novel insights" are. This language needs correcting.

Author Response

Review 2

1) Abstract and conclusion - Inulin shown no overall benefit to ADG, but only a tendency in phase 2 and 3. This is not clearly reflected in the conclusions. Please adequately address these.

Re: Thanks for your suggestion. We have revised this part in the revised manuscript.

2) The authors still need to discuss the biological relevance of a 0.13 ug/L difference in IGF-1. I think they are over interpreting their data and this is reflective by the lack of overall growth differences.

Re: Thanks for your suggestion. We only observed a tendency of increase in the growth performance (i.e ADG), this is probably attributed in part to the elevated serum IGF-1 concentration. We agree with the reviewer that other signaling pathways may participate in this biological process. However, according to the result from statistics, 0.13 ug/L is enough to output a significant result. Other biological pathway may participate in this process. However, we agree with the reviewer’s comment that we may be over interpreting their role. We have changed the description (for instance, the altered IGF-1 may contributed to….) in the text.

3) Conclusion - still not sure on what teh "novel insights" are. This language needs correcting.

Re: Thanks for your suggestion. We have corrected the language in this part. Actually, this is the first report on the influence of inulin on the carcass trait and meat quality in pigs. However, we agree with the reviewers that we should play down the “novel insights” in the manuscript.

Reviewer 3 Report

General comment

The Authors have provided a substantial improvement Into the manuscript. The most of answers, and thus amendments in the text can be accepted, however, the manuscript still needs some work before final acceptance.

Specific comments

Introduction is much more complete now, and in my opinion can be accepted in the present form.

M&M in most of paragraphs also is acceptable, however, I would like to see some more detailed improvements:

Line 89-90. The repetition of days in stages is not necessary in this sentence. The information in line 79 in enough clear. The Authors did not explain the difference in the composition of control and experimental diets. I need only an information about 0.5% of which material was removed to introduce inulin, and how could this change influence the nutrients level (first of all energy an protein content). I still do not fully understand the n values in particular periods of experiment. In the answers the Authors claimed that for statistics of fattening performance all the pigs were used. All, in this situation means for me 18 per group. Meanwhile, in line 192-193 there is an information that pen was used the experimental unit. Additionally, in the description under table 3 there is an information “Values are expressed as the mean of six pigs in each group”. So, there are 3 different information in the same subject. In this situation I cannot accept an answer, because it is different from description in manuscript. I can agree that n=6 can be minimal number for biological and mechanism studies. In the described experiment blood analyses, liver analyses and gene expressions can be classified as mechanism study. This is standard method, that for such an analyses one animal per replicate is chosen, mainly because of costs. But the rest of described analyses are of practical manner and should be made separately on every animal in experiment. Analyses of fattening performance, carcass traits and meat quality are relatively cheap and needs no large labour, and in my opinion minimal n value in such analyses should be 15, optimal 25. I still cannot find justification of decreasing the number of animals in this part of experiment. In the answer the Authors showed classical work of Zybert et al. 2014 as methodological basis of pH analyses. I think they should use this paper as citation in M&M.

Results are much better and much more clear now. The change of charts with tables made the data analysis easier for reader. However, I still think that the Authors should use standard error of the mean separately for each group .

Discussion also seem to be better now. Some additional information made it a little more critical in view of own data, and some trial to interpret data emerged. However, also in this section some amendments are still needed.  

Line 258-260. This is not entirely true. In the own data of the Authors, weaners (pigs during the first stage of experiment 1-32 d) had ADG lower than control, and FCR was also worse. I think that this sentence should be rebuilt, and some trial to interpret this information should be added. Interpretation can be speculative. The discussion is also the place for speculations. Line 263-264. The sentence is repeated twice. Please erase one.

Conclusion is less speculative and more clear now.

Line 313. change “these” with “the”.

Line 314. Erase “the” before beneficial

To summarize, the Authors did good work improving the manuscript substantially, however, in my opinion it still needs some another amendments. I suggest moderate revision.   

Author Response

The Authors have provided a substantial improvement Into the manuscript. The most of answers, and thus amendments in the text can be accepted, however, the manuscript still needs some work before final acceptance.

Re: Thank you. We have made necessary revision based on the reviewer’s comments. All the necessary information has been provided in the revised manuscript.

Specific comments

Introduction is much more complete now, and in my opinion can be accepted in the present form.

Re: Thank you.

M&M in most of paragraphs also is acceptable, however, I would like to see some more detailed improvements:

Line 89-90. The repetition of days in stages is not necessary in this sentence. The information in line 79 in enough clear. The Authors did not explain the difference in the composition of control and experimental diets. I need only an information about 0.5% of which material was removed to introduce inulin, and how could this change influence the nutrients level (first of all energy an protein content). I still do not fully understand the n values in particular periods of experiment. In the answers the Authors claimed that for statistics of fattening performance all the pigs were used. All, in this situation means for me 18 per group. Meanwhile, in line 192-193 there is an information that pen was used the experimental unit. Additionally, in the description under table 3 there is an information “Values are expressed as the mean of six pigs in each group”. So, there are 3 different information in the same subject. In this situation I cannot accept an answer, because it is different from description in manuscript. I can agree that n=6 can be minimal number for biological and mechanism studies. In the described experiment blood analyses, liver analyses and gene expressions can be classified as mechanism study. This is standard method, that for such an analyses one animal per replicate is chosen, mainly because of costs. But the rest of described analyses are of practical manner and should be made separately on every animal in experiment. Analyses of fattening performance, carcass traits and meat quality are relatively cheap and needs no large labour, and in my opinion minimal n value in such analyses should be 15, optimal 25. I still cannot find justification of decreasing the number of animals in this part of experiment. In the answer the Authors showed classical work of Zybert et al. 2014 as methodological basis of pH analyses. I think they should use this paper as citation in M&M.

Re: Thanks for your suggestion, we have added the nutrient level of inulin in Table1 of the revised manuscript. Moreover, we must clarify that 0.5% of which material was replaced by inulin (the corn was replaced) had only few influence on the diet composition. It only serves as an additive. For the statistics of growth performance (i.e. ADG and ADFI) all the pigs were used [each group contains 6 replicates (pen), and each replicate contains 3 pigs] (n=18). However, only one pig (closer to average body weight) from each pen was selected for slaughter (n=6). Six pigs meet the minimal requirement for the statistics (especially for mechanism studies). In china, due to the African swine fever, it is difficult to organize a large-small animal trial. Six replicates (pigs closed to the average body weight) is enough to meet the minimal requirement for statistics.

Results are much better and much more clear now. The change of charts with tables made the data analysis easier for reader. However, I still think that the Authors should use standard error of the mean separately for each group .

Re: Thanks for your suggestion, we have changed the tables in the revised manuscript.

Discussion also seem to be better now. Some additional information made it a little more critical in view of own data, and some trial to interpret data emerged. However, also in this section some amendments are still needed.  

Re: Thanks for your suggestion on our manuscript. We have tried to correct this manuscript based on reviewer’s suggestions. We agree with the reviewer that this is only a preliminary evaluation of the effect of inulin supplementation for animals. If possible we will perform more trials to indicate the mechanisms behind the inulin-mediated responses in the growing-finishing pigs.

Line 258-260. This is not entirely true. In the own data of the Authors, weaners (pigs during the first stage of experiment 1-32 d) had ADG lower than control, and FCR was also worse. I think that this sentence should be rebuilt, and some trial to interpret this information should be added. Interpretation can be speculative. The discussion is also the place for speculations. Line 263-264. The sentence is repeated twice. Please erase one.

Re: Thanks for your suggestion, we have re-written this sentence.

Conclusion is less speculative and clearer now.

Re: Thank you for you suggestion, we have revised this part in the text.

Line 313. change “these” with “the”.

Re: Thanks for your suggestion, we have changed the word in the revised manuscript.

Line 314. Erase “the” before beneficia

Re: Thanks for your suggestion, we have deleted the word in the revised manuscript.
